# Glucosylsphingosine affects mitochondrial function in a neuronal cell model
Valeria Nikolaenko [1], Reddy Vootukuri[1], Simon Eaton [2], Jenny Hällqvist[1], Tomas Baldwin[1], Kevin Mills [1] & Wendy Heywood [1] ✉

Gaucher disease arises from mutations in glucocerebrosidase resulting in accumulation of glucosylceramide, which is deacylated to glucosylsphingosine. Mutations in glucocerebrosidase are the greatest known genetic risk factor for Parkinson's disease. Glucosylsphingosine is a biomarker for Gaucher disease and studies demonstrate its relevance to disease pathology, yet the mechanisms of its toxicity remain little understood. Using proteomics, we show that incubation of SH-Sy5y cells with glucosylsphingosine at physiological plasma concentrations observed in moderate/ severe Gaucher disease negatively effects the TCA cycle, mitochondrial function, glycolysis and protein ubiquitination. Functional analyses confirmed that glucosylsphingosine reduces ATP production, elicits oxidative stress and an increase of glycolysis. Analyses of ubiquitinated proteins and lipid-binding studies demonstrated that glucosylsphingosine has binding affinity for tubulin alpha and induced a specific increase of ubiquitination of α and β tubulins. In conclusion, supranormal levels of glucosylsphingosine affect cellular energy metabolism which may contribute to the pathology in Gaucher disease.

Homozygous mutations in the *GBA* gene that encodes for the lysosomal acid beta glucosidase (GBA1) cause the most common lysosomal storage disorder Gaucher disease (GD)[1].Gaucher disease is a heterogeneous disease and is categorised by the absence (type I) or presence (types II and III) of the central nervous system involvement with GD II being the most severe[2]. GBA1 breaks down the glycosphingolipid glucosylceramide. The downstream consequence of glucosylceramide accumulation is its modification to glucosylsphingosine (GlcSph), where the fatty acid moiety is removed from the accumulating glucosylceramide by the lysosomal enzyme acid ceramidase[3]. Plasma levels of GlcSph strongly correlate with disease burden and response to treatment, with highest levels observed in the brain of neuronopathic GD[4]. GlcSph is considered a superior biomarker than the accepted GD biomarker chitotriosidase. However, chitotriosidase is not specific to GD as it can be elevated in other diseases and cannot be measured in some populations who carry a 24–base pair duplication in the chitotriosidase gene[5–10]. Exogenous GlcSph causes the same haematological and visceral changes in mice as those observed in GD patients, while endogenous GlcSph interferes with lysosomal biogenesis, promotes a-synuclein aggregation and toxicity, directly correlates with a-synuclein levels in Parkinsonian brain, is toxic to cholinergic neurons and causes loss of dopaminergic neurons[11–14].

Decreased GBA1 activity results in mitochondrial dysfunction with loss of mitochondrial membrane potential, reduced ATP, increased oxidative stress and fragmented mitochondria and is observed in GD fibroblasts, GD cell and animal models and is a well-established feature of Parkinson's disease (PD)[15–17]. Despite the established connection between GlcSph and some of the common pathology of GD and PD, the exact mechanism of GlcSph cellular toxicity and effect on mitochondria is not clear. We therefore aimed to evaluate the effect of GlcSph on mitochondria at the two physiological concentrations observed in moderate (20 ng/mL) and severe (200 ng/mL) GD phenotypes using a neuronal SH-Sy5y cell model. To determine the specific effect of GlcSph we have used another disease associated lyso-glycosphingolipid, lyso-Gb3, as a disease control. Lyso-Gb3 differs from GlcSph by a trihexoside moiety with two galactose and a glucose molecule instead of a single glucose attached to the ceramide. Lyso-Gb3 is associated with Fabry disease which has a very different clinical phenotype to GD and no neurological involvement. Our previous work has shown it also has its own specific effect on the cell[18]. The work presented here expands on the effect of GlcSph on the cell's metabolic activity and how this could cause the associated metabolic phenotype in GD and potentially shed light on the role of GlcSph in PD.

[1]Translational Mass Spectrometry Research Group, Genetics & Genomic Medicine Dept., UCL Institute of Child Health, London, WC1N 1EH, UK. [2]Developmental Biology and Cancer Programme, UCL Great Ormond Street Institute of Child Health, London, UK. ✉e-mail: wendy.heywood@ucl.ac.uk

## Results

GlcSph entry into cells from culture medium was confirmed by UPLS-MS/MS and was proportionately increased with increasing concentrations of GlcSph supplementation (Supplementary Fig. S1).

### GlcSph exposure causes proteomic changes in metabolic pathways in SH-Sy5y cells

Brain tissue concentrations of GlcSph varies from 0.4 to 4.7 mg/kg among GD I – III cases and are 100–1000-fold increased compared to the normal brain, and plasma levels reach 250 ng/mL but depend on the type and the severity of *GBA1* mutation and residual enzyme activity[19,20]. To gain insights into molecular mechanisms of GlcSph toxicity we incubated SH-Sy5y cells with GlcSph at 20 ng/mL, which is observed in mildly affected GD patients, for 24 h to test for the acute response on the cellular proteome. Cells were also incubated with GlcSph at 200 ng/mL for 72 h to test for a longer-term effect. The disease control lyso-Gb3 was used at the same concentrations. DMSO was used as a vehicle control.

As previously reported both GlcSph and lyso-Gb3 affect the cellular proteome, however, GlcSph exposure resulted in a bigger change in proteome expression than lyso-Gb3. At both low and high doses GlcSph affected approximately 10% and 16% of the proteome respectively *versus* 5.8% and 12.4% compared to lyso-Gb3 (Fig. 1a). While the 20 ng/mL treatment resulted in 21% reduction in protein level for both lyso-lipids, the larger dose of 200 ng/mL led to a greater increase in the number of down-regulated proteins for GlcSph (29% vs 19%) (Fig. 1b). To gain insights into biological pathways affected by the lyso-lipids we performed a bioinformatics pathway analysis looking at the changes in protein expression after treatment using Ingenuity Pathway Analysis tool (Qiagen). The analysis suggested that GlcSph mainly affects metabolic pathways whilst lyso-Gb3 affects signalling pathways (Supplementary Tables S1–S4). At the low dose of 20 ng/mL GlcSph affected the TCA cycle the most, followed by RAN

signalling (involved in metabolic regulation and dependent on the energy status of the cell), mitochondrial dysfunction and sirtuin signalling (Fig. 1c).

In the high dose of 200 ng/mL and similarly to the low dose mitochondrial dysfunction and sirtuin signalling pathways remained disturbed, however, they were superseded by tRNA charging (involved in protein translation) and protein ubiquitination pathways. In addition. glycolysis was also affected (Fig. 1d). These data indicate that GlcSph affects cellular metabolism and with increased dose likely starts to affect protein folding and degradation.

As mitochondrial dysfunction was a pathway specifically affected by GlcSph exposure in both doses a further analysis was performed looking at mitochondrial protein expression alone and their relation to key mitochondrial processes to get a deeper understanding. Figure 1e shows the percent of significantly affected proteins in each process defined by the total identified proteins assigned to that process. The low dose of 20 ng/mL appears to affect the TCA cycle, complex I, IV and V but the high 200 ng/mL dose also affects complex III and starts to affect acetyl CoA metabolism and fatty acid degradation.

### GlcSph reduces ATP production rate and induces metabolic shift towards glycolysis

To explore further the effect of GlcSph on energy metabolism we investigated ATP production and glycolytic rates in live SH-Sy5y cells exposed to the two doses of GlcSph using Seahorse technology (Agilent, USA). This technology simultaneously measures oxygen consumption and extracellular acidification rates to infer the two major energy producing processes – oxidative phosphorylation (OXPHOS) and glycolysis. Cells were incubated with GlcSph at 20 ng/mL and 200 ng/mL for 72 h and assessed for ATP production and glycolytic rates (basal measurement). To examine whether GlcSph also affects cellular capacity to switch between alternative

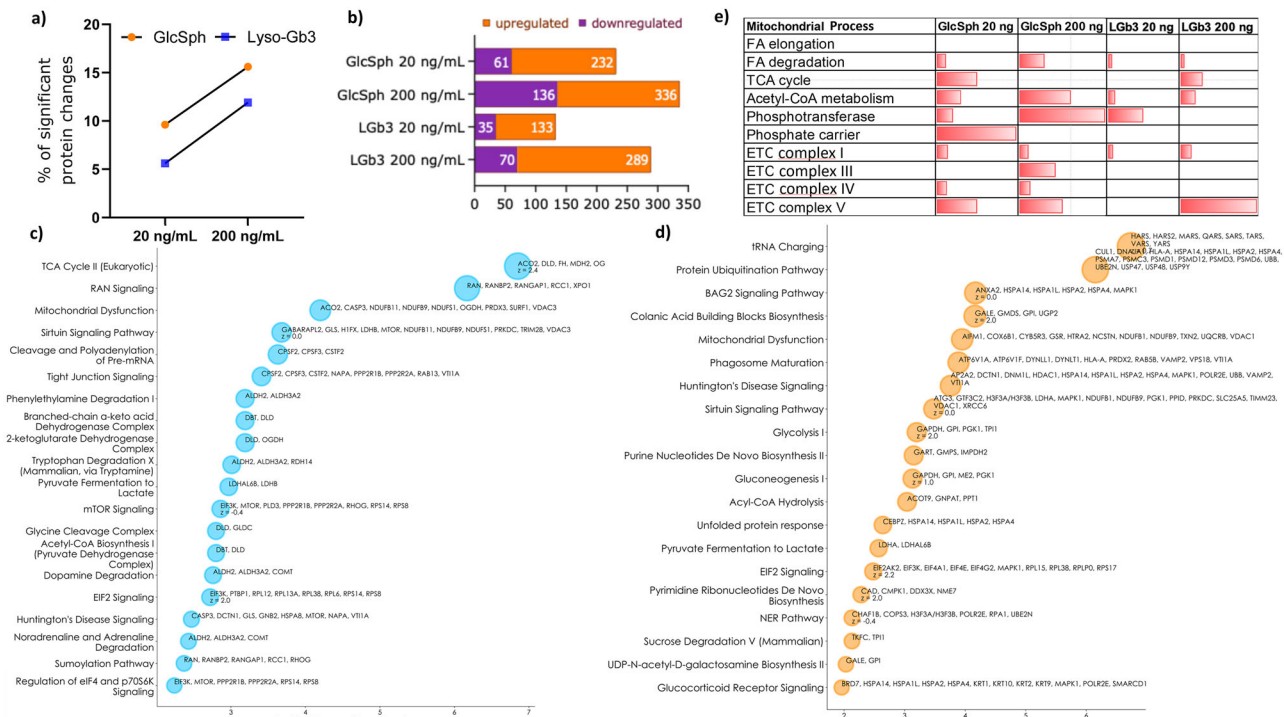

**Fig. 1 | Exposure to GlcSph and lyso-Gb3 at 20 ng/mL and 200 ng/mL significantly alters protein expression in the SH-Sy5y cell line. a** Percent of significant (*p* < 0.05) protein changes of the total proteome by the lysolipid and dose. **b** The number of significant up-/down-regulated proteins. Top 15 significantly affected cellular pathways by GlcSph at **c** 20 ng/mL and **d** 200 ng/mL. Circle radii represent

the negative log10 of the pathway enrichment *p*-value < 0.05 and the proteins affected in the pathway. z-scores > 0 indicate activation while z-scores < 0 indicate deactivation of the pathway. **e** Significantly affected mitochondrial proteins by each lyso-lipid and dose. Source data are provided as a Supplementary Data file.

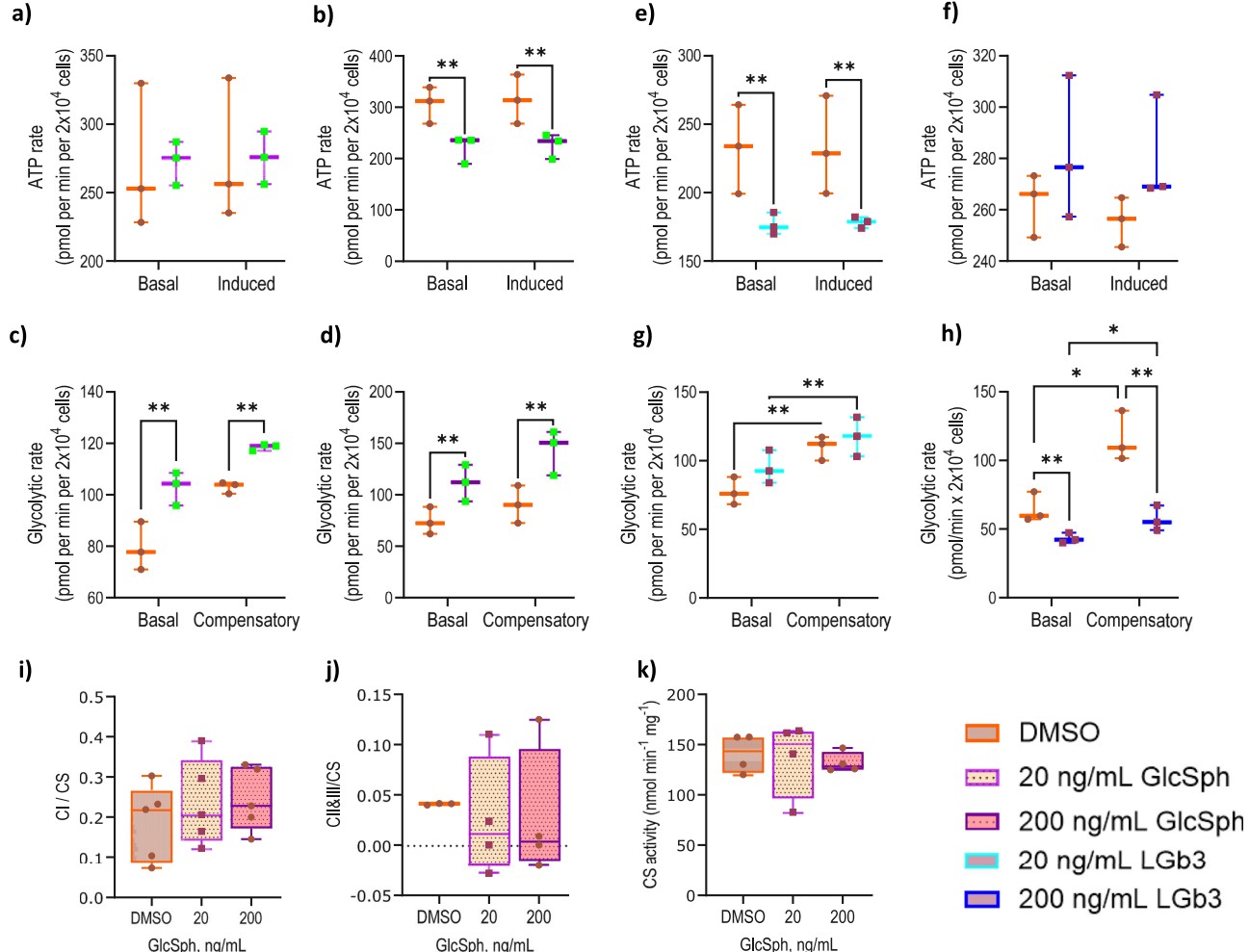

**Fig. 2 | The effect of GlcSph on energy metabolism and mitochondrial function in live SH-Sy5y cells.** ATP production rate in 20 ng/mL (**a**, **e**) and 200 ng/mL (**b**, **f**) of GlcSph and lyso-Gb3 exposures respectively. Glycolytic rate in 20 ng/mL (**c**, **g**) and 200 ng/mL (**d**, **h**) of GlcSph and lysoGb3 exposures respectively. Induced measurement denotes inhibition of pyruvate entry into mitochondria in the ATP analyses, compensatory measurement denotes total OXPHOS inhibition by competitive inhibition of hexokinase by deoxyglucose in glycolysis analyses. DMSO was used as a vehicle control. Box plots show mean ±25% (boxes) and max. and min. values (whiskers) ($n = 3$ for (**a**–**h**)). Statistical significance was determined by a 2-way ANOVA followed by a Sidak's post-hoc test. The activity of **i** complex I ($n = 5$), **j** complex II&III ($n = 4$), and **k** citrate synthase ($n = 4$) was unaffected. Statistical significance was determined using a Kruskal–Wallis test with a Dunn's post-hoc test. $*p < 0.5$, $**p < 0.01$. Source data are provided as a Supplementary Data file.

pathways of ATP production, a mitochondrial pyruvate carrier inhibitor was used (induced measurement) for the ATP production rate.

The analyses shown in Fig. 2 demonstrated that cellular ATP production rate was not affected by GlcSph exposure at 20 ng/mL with or without mitochondrial pyruvate carrier inhibition (Fig. 2a). This suggests at the low 20 ng/mL dose GlcSph did not affect the mitochondrial OXPHOS and the cell's ability to switch fuels. However, that was not the case for the high dose of 200 ng/mL, which led to a significant 28% drop in ATP production rate ($p = 0.0025$) and the immediate cellular response to the inhibition of the pyruvate entry into the mitochondria ($p = 0.0025$) (Fig. 2b). This was contrasted with the glycolytic rate, which was significantly increased at both dose exposures (Fig. 2c, d). Basal glycolysis under GlcSph exposure increased by 45% ($p = 0.0071$) in response to the low dose, which also affected the cell's ability to compensate for the total inhibition of mitochondrial OXPHOS (compensatory rate) and led only to a 16% increase in glycolytic rate in treated cells ($p = 0.015$) vs 37% for the DMSO control cells ($p = 0.015$). At the high dose GlcSph led to an even higher (50%) increase in basal glycolysis ($p = 0.0066$), while compensatory glycolysis was increased by 58% ($p = 0.0066$). Furthermore, GlcSph exposed cells at the

high dose were able to upregulate glycolysis more than the DMSO control cells (28% vs 21%) with the total inhibition of OXPHOS (Fig. 2d).

In contrast, the disease control treatment with lyso-Gb3, at the low 20 ng/mL dose led to a significant 24% reduction in both basal ($p = 0.0053$) and induced ($p = 0.0053$) ATP production rates (Fig. 2e) and suggests that lyso-Gb3 affects the mitochondrial ATP production and the immediate cellular response to the mitochondrial pyruvate entry inhibition. However, this was not the case when the cells were exposed to the high 200 ng/mL dose of lyso-Gb3, which demonstrated no effect on ATP production rate (Fig. 2f). When looking at the glycolytic rate we observe a different picture. Lyso-Gb3 did not affect glycolysis at the low dose compared to the DMSO control, although a 31% reduction was observed at the 200 ng/mL dose. However, with the total inhibition of mitochondrial OXPHOS (compensatory measurement) lyso-Gb3 treated cells at the 20 ng/mL dose were able to upregulate glycolysis only by 24% vs 41% compared to DMSO control (Fig. 2g). The same was observed at the high 200 ng/mL dose where lyso-Gb3 treated cells upregulated compensatory glycolysis only by 40% vs 79% observed in the DMSO control (Fig. 2h). This implies that lyso-Gb3 exposure impaired the cell's ability to compensate via glycolysis for the inhibited OXPHOS in opposite to GlcSph exposed cells.

**Fig. 3 | The effect of GlcSph on cellular stress in SH-Sy5y cells.** GlcSph exposure does not affect **a** adenylate energy charge (*n* = 4), and the activity of **b** cytosolic and **c** mitochondrial malate dehydrogenases (*n* = 6) in SH-Sy5y cells. **d** GlcSph exposure elicits an increase in the activity of glutathione reductase at 200 ng/mL (*n* = 9). DMSO was used as a vehicle control. Box plots show mean ±25% (boxes) and max. amd min. values (whiskers). Statistical significance was determined using a Kruskal–Wallis test with a Dunn's post-hoc test. *$p < 0.5$. Source data are provided as a Supplementary Data file.

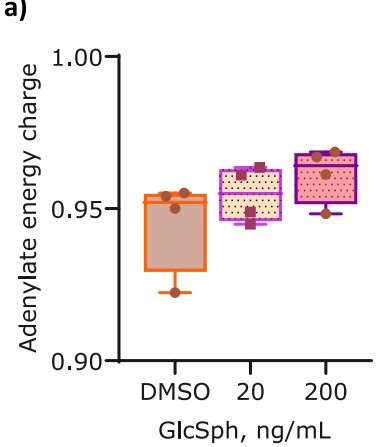

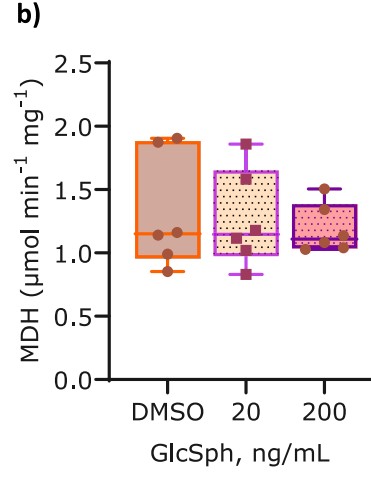

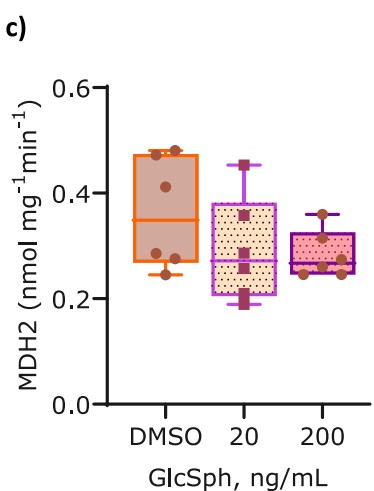

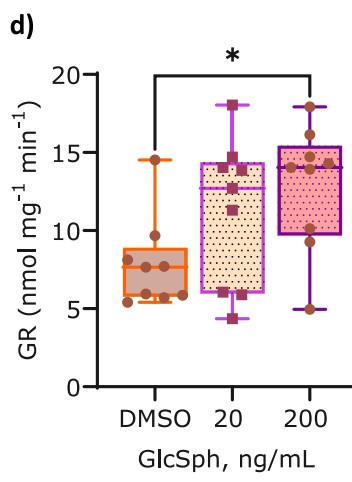

## GlcSph exposure does not affect the activity of mitochondrial complexes I – III

As proteomic profiling of SH-Sy5y cells exposed to GlcSph at both concentrations demonstrated dysregulation of ETC complex proteins the activity of complex I, which is impaired in both GD and PD, was investigated further[21–23]. Citrate synthase being the first rate limiting enzyme of the TCA cycle is often used as a marker of mitochondrial content and its activity was also measured and used for subsequent normalisation of the activity of complex I and II/III[24]. The analysis demonstrated that citrate synthase activity itself was unaffected and stable. Overall, there was no significant difference in the activity of complex I-III between GlcSph treated and control cells (Fig. 2i, j). However, both analyses demonstrated highly variable activity between biological replicates particularly for complex II/III which could not be explained. Technical causes were ruled out as the same effect was observed when repeated, with stable activity of citrate synthase observed in the DMSO control group (Fig. 2k).

## GlcSph exposure does not affect the cell's ability to maintain its cellular energy charge

To ascertain how the reduction in ATP production rate in GlcSph exposure might affect the bioenergetic status of the cell we measured adenylate energy charge using HPLC-chemical detection. No difference in adenylate energy charge was observed in both GlcSph exposures (Fig. 3a) and suggests that despite the negative effect of GlcSph on the ATP production rate the cell is still able to maintain higher cellular ATP levels relative to ADP and AMP.

This is likely compensated by the upregulated glycolysis observed (Fig. 2c, d).

## GlcSph exposure does not affect malate-aspartate shuttle and malate dehydrogenase activity

One way the cell's energy could be affected is through NAD+ availability. Proteomics analysis indicated altered expression of mitochondrial malate dehydrogenase, therefore we investigated the functional effect of GlcSph on malate dehydrogenase activity.

We observed no difference in the activity of cytosolic malate dehydrogenase between GlcSph treated and control cells (Fig. 3b), which suggests the availability of NADH for its activity was not altered. Although a diminished trend (approx. 18%) was observed for the activity of mitochondrial malate dehydrogenase between GlcSph treated and control cells this decrease was not significant (Fig. 3c). These results suggest that GlcSph exposure does not affect malate-aspartate shuttle and regeneration of NADH in mitochondria and its consumption in the cytosol.

## GlcSph exposure induces cellular stress that is compensated in the short term

Mitochondrial dysfunction and reduced ATP production indicate potential for cellular stress caused by GlcSph. Proteomic analysis indicated upregulation of glutathione reductase (Supplementary Table S5), therefore its activity was assessed. Glutathione reductase is an important enzyme in the glutathione cycle – the main cellular antioxidant defence system.

**Table 1 | The fourteen proteins that bound GlcSph**

| Gene name | Protein | Sub cellular location |
|---|---|---|
| TUBA3C | Tubulin alpha-3C chain | Cytosolic, |
| ENO1 | Alpha enolase | Cytosolic, Cell membrane |
| HCD2 | 3-hydroxyacyl-CoA dehydrogenase type-2 | Mitochondria |
| LYPLA1 | Acyl protein thioesterase 1 | Cytosolic, CM, NM, ER |
| SLC25A5 | ADP/ATP translocase 2 | Mitochondria IM |
| SLC25A6 | ADP/ATP translocase 3 | Mitochondria IM |
| LDHB | L-lactase dehydrogenase beta chain | Cytosolic, Mitochondria IM |
| ACTB | Actin cytoplasmic | Cytosolic, cytoskeleton |
| PHB1 | Prohibitin | Cytosolic, Mitochondria IM, Nucleus |
| HADHA | Trifunctional enzyme subunit alpha | Mitochondria IM |
| HNRNPU | Heterogeneous nuclear ribonuclear protein U | Cytosolic, Nucleus, Cytoskeleton |
| PCB1 | Poly(r)C-binding protein 1 | Cytosolic, Nucleus |
| PCB2 | Poly(r)C-binding protein 2 | Cytosolic, Nucleus |
| H2B1K | Histone H2B type K-1 | Nucleus |

*CM* Cell membrane, *IM* inner membrane.

Glutathione is reduced by glutathione reductase using NADPH as an electron donor and in its reduced form (GSH) assists in clearance of hydrogen peroxide and lipid peroxides. GSH also donates electrons to ER resident protein isomerases required for proper protein folding.

The analysis demonstrated that the activity of glutathione reductase was increased in both low and high dose GlcSph exposure although this increase was only statistically significant at the high 200 ng/mL dose ($p = 0.041$) (Fig. 3d). This result suggests there may be an elevation in reactive oxygen species in GlcSph treated cells.

### GlcSph specifically binds and causes ubiquitination of tubulin species

Proteomic analysis indicated protein ubiquitination was a pathway upregulated by exposure to both lyso-lipids. We previously reported, that lyso-Gb3 elicited a greater effect on this pathway, however, the proteins affected by either lipid differ indicating specific lyso-lipid effects[18]. To explore which proteins have a high affinity for GlcSph and vice versa, and potentially are prone to ubiquitination we immobilised GlcSph onto magnetic beads and incubated with SH-Sy5y cell lysates. Of note, immobilisation of lyso-lipids is expected to obscure their free amine group as it reacts with the epoxy group on magnetic beads. This may have affected the proteins immobilised GlcSph bound to, and is a limitation of this study. However, the effects on the cell proteome and associated disease phenotypes of the two lyso-lipids are so different we suspect that the differences are largely caused by the different sugar moieties and not just the common amine group that both lyso-lipids have. Hence why a comparison with blank beads and lyso-Gb3 was considered the best way to look for specific effects of GlcSph. Cellular proteins that were ubiquitinated after GlcSph and lyso-Gb3 exposure were isolated using an antibody that recognised mono- and polyubiquitin chains on proteins and captured proteins were analysed by label free proteomics. The analysis demonstrated that GlcSph bound 14 proteins listed in Table 1, of which only two proteins, namely tubulin alpha 3C and alpha enolase bound exclusively. We further assessed if GlcSph binding to alpha tubulin and alpha enolase may affect their function and cause their ubiquitination. The ubiquitinated immunoprecipitation protein analysis demonstrated that out of the 54 ubiquitinated proteins observed exclusively in the GlcSph exposure both alpha and beta tubulins were the most abundant (Supplementary

Table S6, Fig. 4a), which was different to the lyso-Gb3 exposure (Fig. 4b). Tubulins were also more ubiquitinated in GlcSph treated cells compared to the DMSO control and lyso-Gb3 treatment, particularly the core tubulins of microtubules TUBB (6-fold), TUBB3 (7-fold) and TUBA1B (4-fold) vs control (Fig. 4c) thereby supporting proteomic observations that also indicated dysregulation of tubulin species in GlcSph treated cells (Fig. 4d). These results suggest that GlcSph exposure may specifically render tubulin species less functional or difficult to fold, which could subsequently affect microtubule dynamics and neuronal transport. Gene ontology analysis of the 54 ubiquitinated proteins exclusively observed in the GlcSph exposure also demonstrated enrichment in the ubiquitination of tubulins, chaperones of the 70 kDa family and ATP binding proteins (Table 2) indicating a possible problem with protein synthesis and folding.

With this observation we investigated the possibility of GlcSph potential disruption of tubulin polymerisation, which would disrupt microtubule assembly. Tubulin polymerisation reaction in vitro was performed using a fluorescence-based kit (Cytoskeleton, BK011P). The analysis demonstrated GlcSph had no effect on any of the phases of microtubule formation (nucleation, growth or equilibrium) at the three concentrations tested (Fig. 4e). This suggests that GlcSph exposure may increase tubulin turnover rather than affect microtubule formation/assembly.

The assessment of alpha enolase activity demonstrated no differences between GlcSph treated and control cells and suggests the interaction may be transient (Supplementary Fig. S2).

## Discussion

*GBA1* mutations represent the greatest known genetic risk factor for developing PD[25], whose hallmarks are mitochondrial dysfunction, aggregation and toxicity of a-synuclein protein and loss of dopaminergic neurons[2][6]. The activity of the GBA1 enzyme in the brain of PD patients is also reduced irrespective of the *GBA1* mutation status[27]. Almost undetectable in healthy cells GlcSph accumulates in both GBA1 carriers and GD patients and has been shown to promote a-synuclein aggregation and toxicity[13], and loss of dopaminergic neurons[28]. We sought to investigate the effect of GlcSph on mitochondrial function. Proteomic analysis of SH-Sy5y cells exposed to GlcSph demonstrated altered metabolism even at the lower mild levels observed in GD phenotypes. These changes, shown in Fig. 1c-d, indicated activated TCA cycle (z-score = 2.4) and mitochondrial dysfunction. The higher level typically observed in severe GD phenotypes indicated activated glycolysis (z-score = 2) and mitochondrial dysfunction. We have confirmed this specific effect of GlcSph on cell metabolism using functional assays to demonstrate changes in ATP production and glycolytic rates in live cell analyses further to support our initial proteomic findings. Our findings show that glycolysis can rescue reduced ATP production rate at the low dose but not adequately at the high dose. As suggested by the proteomic analysis (Fig. 1e) the TCA cycle may also be contributing to the maintenance of the ATP production rate and warrants further investigation. This contrasted with the disease control lyso-lipid exposure to lyso-Gb3, which demonstrated a different effect on cell metabolism where a 24% reduction in the ATP production rate was observed at the low 20 ng/mL dose (Fig. 2e). Curiously, no effect at the higher dose was observed indicating a low dose of lyso-Gb3 may be more potent on cell function and suggests there may be some cellular adaptation long term that compensates. We previously observed this effect of lyso-Gb3 on protein ubiquitination[18]. Intriguingly, lyso-Gb3 had an opposite effect on the glycolytic rate compared to GlcSph where a 31% reduction was observed during OXPHOS inhibition and indicates the cell was not able to compensate adequately (Fig. 2g, h). Furthermore, compensatory glycolysis for the inhibited OXPHOS was also hampered in lyso-Gb3 exposed cells compared to control. These results suggest that each lyso-lipid has a specific and opposite effect on the ATP production and glycolytic rates whereby GlcSph induces a shift toward glycolysis while lyso-Gb3 does not. This could be due to the significant differences in the lipid structures (lyso-Gb3 is bulkier and more hydrophilic), their ability to cross cellular membranes, be transported in the cell and accumulate in different cellular compartments. They also demonstrate

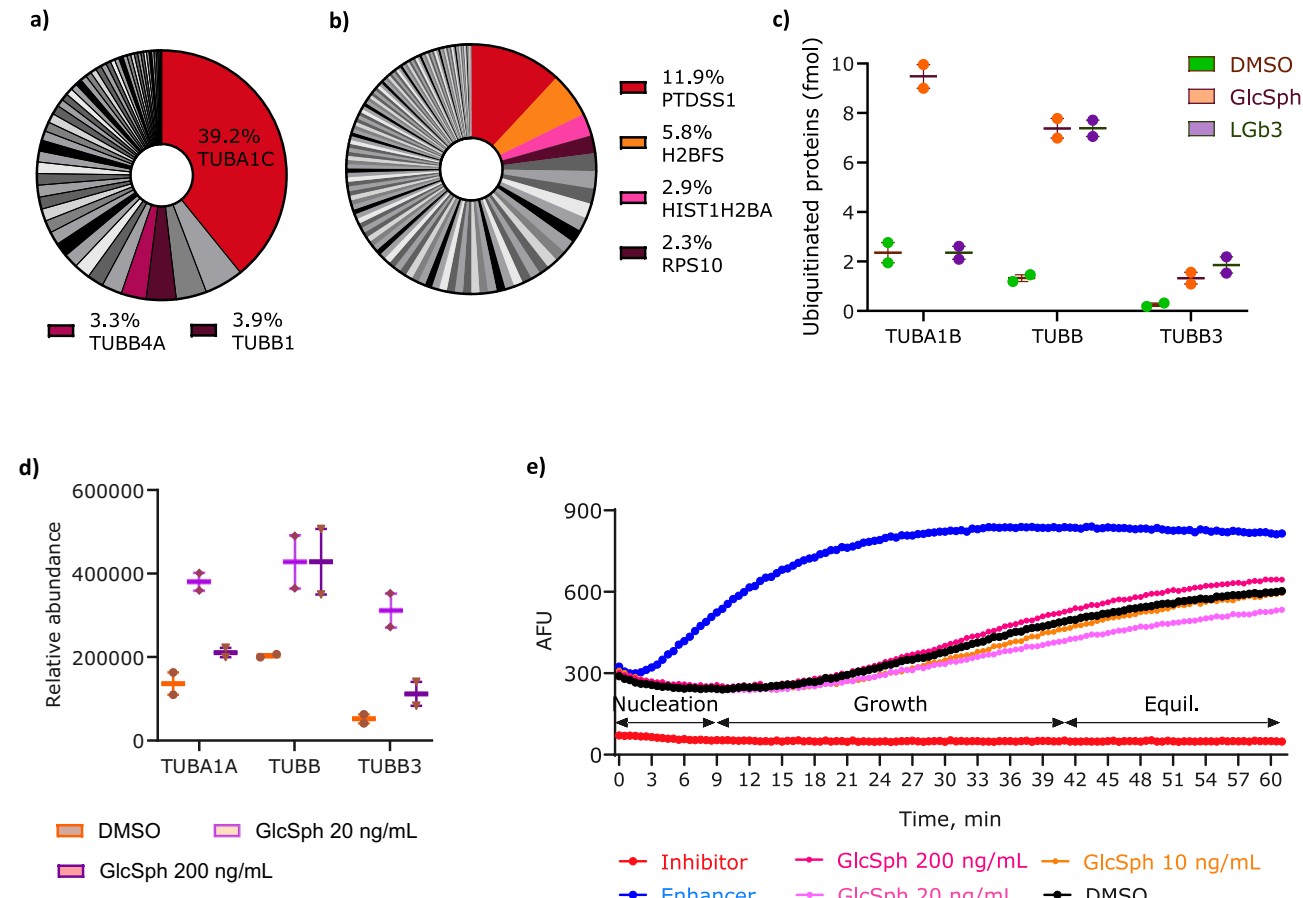

**Fig. 4 | GlcSph exposure increases ubiquitination of tubulin species in SH-Sy5y cells. a** Pie chart showing tubulin species are ubiquitinated the most among the proteins exclusive to GlcSph exposure in contrast to **b** ubiquitinated proteins exclusive to lyso-Gb3 exposure. **c** Tubulin species observed to be differentially ubiquitinated in GlcSph and lyso-Gb3 treated cells vs control. **d** Proteomic analysis demonstrating altered abundance of tubulin species in GlcSph treated cells. Box plots show mean ±25% (boxes) and max. and min. values (whiskers) (n = 2). **e** Polymerisation assay demonstrating no effect of GlcSph at either 20 ng/mL or 200 ng/mL on any of the polymerisation phases. Source data are provided as a Supplementary Data file.

**Table 2 | Summary of the GO analyses of the ubiquitinated proteins unique to GlcSph exposure**

| PANTHER cellular component | Homo sapiens (REF) | Found | Expected | Fold enrichment | Raw p-value | FDR |
|---|---|---|---|---|---|---|
| Intermediate filament cytoskeleton | 51 | 5 | 0.13 | 39.58 | 2.62E−07 | 3.34E−05 |
| Cytoskeleton | 751 | 9 | 1.86 | 4.84 | 9.01E−05 | 2.07E−03 |
| Ribosomal subunit | 80 | 4 | 0.20 | 20.19 | 5.51E−05 | 1.29E−02 |
| PANTHER molecular function | | | | | | |
| Misfolded protein binding | 16 | 3 | 0.4 | 75.69 | 1.34E−05 | 4.57E−04 |
| Heat shock protein binding | 33 | 3 | 0.8 | 36.7 | 9.58E−05 | 2.91E−03 |
| ATP binding | 41 | 4 | 0.10 | 39.39 | 4.57E−06 | 1.66E−04 |
| PANTHER protein class | | | | | | |
| Hsp70 chaperone | 15 | 3 | 0.4 | 80.74 | 1.13E−05 | 2.21E−03 |
| Tubulin | 24 | 4 | 0.6 | 50.46 | 3.99E−05 | 3.91E−03 |
| Ribosomal protein | 177 | 5 | 0.44 | 11.40 | 8.53E−05 | 5.57E−03 |
| PANTHER pathway | | | | | | |
| Serine glycine biosynthesis | 6 | 2 | 0.1 | >100 | 1.66E−04 | 3.79E−03 |
| Parkinson disease | 101 | 4 | 0.25 | 15.99 | 1.31E−04 | 4.21E−03 |

that GlcSph exposure could contribute to the metabolic changes observed in GD and possibly GBA associated PD. Decreased cellular ATP levels are observed in fibroblasts of patients with familial PD[29], PD patients' brains[21], and platelets[22], but this effect has been attributed to reduced mitochondrial

complex I activity. Reduced ATP levels have also been observed in a neuronopathic GD mouse model[30] and peripheral blood mononuclear cells of GD patients[31], with reduced complex I-III activity in the brains of Gba null mouse model of neuronopathic GD[23]. We attempted to confirm if complex I

and II/III activity could be the cause for reduced ATP production demonstrated in this study, and increased stress indicated by the proteomic analysis, however, GlcSph exposure resulted in highly variable activity of complex II/III compared to stable activity of the DMSO control (Fig. 2i, j). The same effect was observed on several occasions when experiment was repeated, so whilst we cannot confirm a significant effect on complex II/III activity it does appear there is some variable effect which could be due to how the lipid behaves/transports within the cell or due to the cell type used in this study. Thus whilst we demonstrate metabolic reprogramming with the shift toward glycolysis and reduction in the ATP production rate that is associated with loss of complex I in dopaminergic neurons in mice[32], further investigations are necessary to determine whether GlcSph truly does not affect the activity of complexes I-III in other, more vulnerable cell types such as dopaminergic neurons that are more reliant on OXPHOS[28]. Increased glycolysis as observed in this study has been suggested to be not only a compensatory but a protective mechanism in various neurodegenerative diseases including PD[33] and can protect against a-synuclein proteotoxic effect[34] and cellular stress via enhanced chaperone activity of Hsp90[35].

Despite the negative effect of GlcSph on the ATP production rate our findings show the cell appears to be able to maintain its cellular energy charge, i.e. the ability of the cell to maintain its ATP levels higher than those of ADP and AMP as this was unaffected (Fig. 3a). This may be due to an increased expression of the mitochondrial creatine and adenylate kinases, voltage dependent anion channels (VDACs) and phosphate carrier proteins as was suggested by the proteomic analysis (Supplementary Table S5) and whose job is to facilitate cytosolic ATP availability quickly and maintain global and local ATP/ADP ratios. These data also suggest a well-controlled balance between the ATP producing and ATP consuming reactions, possibly due to switching off ATP consuming processes such as protein synthesis, folding and degradation and increase in catabolic processes such as amino and fatty acid breakdown. Incidentally, pathway analysis demonstrated tRNA charging and protein ubiquitination pathways as the most significantly altered in the 200 ng/mL dose exposure (Fig. 1e). Gene ontology analysis also indicated misfolded and heat shock protein binding as enriched functions together with heat shock proteins as an enriched class (Table 2) thereby indicating altered protein homeostasis and adding weight to this hypothesis.

In the attempt to ascertain what caused the reduction in the ATP production rate we probed the malate-aspartate shuttle that maintains mitochondrial NAD and cytosolic NADH availability and tested the activity of both cytosolic and mitochondrial malate dehydrogenases involved in the NADH oxidation in the cytosol and its regeneration in the mitochondria respectively. However, the analyses demonstrated no effect of GlcSph exposure on the activity of either enzyme (Fig. 3b, c). Whilst only two proteins specifically bound GlcSph we did observe proteins that bound both lyso-lipids (Table 1), six of which were localised to the mitochondria including the inner membrane. Two of these proteins were ADP/ATP translocase 2 & 3 which facilitate ADP/ATP transport in and out of the mitochondria and when disrupted can lead to impaired OXPHOS. Proteomic analysis indicated upregulation of both translocases in GlcSph exposure but only at the higher dose the increase was significant (Supplementary Table S5). This increase may be due to GlcSph binding and rendering the translocase less functional. SLC25A5 is also a necessary component for parkin-mediated mitophagy[36] and loss or reduction of its activity may also lead to mitochondrial dysfunction in GlcSph exposure as suggested by the pathway analysis (Fig. 1d, e). Whilst the two lyso-lipids bind to ADP/ATP transporters they may have unique, specific effects thus affecting OXPHOS in different ways and this observation warrants further investigation.

Another specific effect observed from GlcSph exposure was the binding of tubulin. We determined that GlcSph does not affect microtubule polymerisation in vitro but likely affects tubulin turnover as it elicited greater levels of ubiquitination of tubulins compared to lyso-Gb3 (Fig. 4a, b). Tubulin is known to associate with VDACs and modulate their permeability thereby affecting ADP/ATP exchange[37]. Upregulation of VDACs as

suggested by the proteomics analysis (Supplementary Table S5) may indicate the cells attempt to increase ADP/ATP exchange. Increased ubiquitination of tubulins indicating their increased turnover and thus altered modulation of VDACs further supports this observation. Microtubule disruption that may result from changes in tubulin turnover can affect many cellular processes and in GD animal models altered lysosome distribution is attributed to cytoskeletal disruption, which occurs in the early stages of disease[38]. Altered lysosomal distribution can impair lysosome function and lead to neurodegenerative disease[39]. Changes in tubulin can cause trafficking defects and lead to adult onset movement disorder in mice[40], disrupted mitochondrial transport in human cells[41], and even cause PD[42]. Changes in TUBA1C, the most abundant ubiquitinated protein unique to GlcSph dataset in this study (Fig. 4a), have been demonstrated in rotenone-induced rat PD model[43], highlighting the toxic role GlcSph may play in GBA related PD.

A limitation of our study is understanding the combined effect of the GlcCer accumulation that also occurs in GBA1 mutant cells. Our work clearly defines the specific effect of GlcSph but the effect of GlcCer accumulation could also contribute and exacerbate some of the findings we describe here and observed in GD. Previous work has addressed both lipids using a double knock out zebrafish model for Gba1 with and without acid ceramidase knocked out[28] This study showed that accumulation of these two lipids does lead to specific effects. GlcCer accumulation was found to be attributed more to neuroinflammation and the absence of GlcSph in Gba1 mutants did lead to reduced disease severity with increased lifespan. An interesting finding was the loss of dopaminergic neurons, which was slowed down in the double knock out mutants, confirming GlcSph effect on vulnerable neuronal cells over time. Our findings of how GlcSph specifically affects mitochondria likely explains why these high energy demanding dopaminergic neurons are affected more by GlcSph than GlcCer accumulation.

In summary, our data show that GlcSph is neurotoxic and its mechanism may be due to altering pathways responsible for ATP production and concomitant increase in oxidative stress. While the cell is able to maintain its cellular charge and compensate by upregulating glycolysis, a rapid but inefficient way of making ATP, this may be sufficient in a short-term situation, which we can only study in this system. However, these cellular deficiencies and a reduced capacity of neuroprotective mechanisms with age or in a long-term chronic exposure may eventually lead to damage of vulnerable cells such as neurons and potentially answer some of the questions regarding the disease mechanisms associated with GD and GBA related PD.

Supplementary information accompanies the manuscript on the Communications Biology website http://www.nature.com/commsbio.

## Methods

### Cell culture

SH-Sy5y cells were purchased from the European Collection of Cell Cultures (Public Health England, UK). Cells used in this study were between passage 8 and 15. Cells were incubated in DMEM/F12 (1:1) supplemented with 10% human serum (Sigma, UK) due to high level of endogenous glucosylsphingosine present in standard 10% FBS supplemented culture medium. Cell viability was assessed using AlamarBlue™ HS cell viability reagent (Invitrogen, Thermo Fisher) as per manufacturer's instructions. Cells were plated in 6 well plates at the density of $3 \times 10^5$ cells/well and treated with lyso-Gb3/GlcSph (Avanti lipids) dissolved in DMSO or vehicle for 24 h at the 20 ng/mL dose and 72 h at the 200 ng/mL dose. Cells were harvested using trypsin and centrifuged at $500 \times g$ for 5 min, and further washed with PBS.

### Label Free proteomics and deep proteomic phenotyping

Pelleted cells were lysed in PBS by three freeze-thaw cycles using 37 °C bath and dry ice. Cell lysate was ice-cold acetone precipitated at a volume of 1:3 and left at −20 °C for 4 h, then centrifuged to pellet precipitated proteins. The supernatant was removed and the protein pellet air dried. Precipitated

protein was digested as described previously[44]. Briefly, protein was solubilised by the addition of 20 μL of 100 mM Tris buffer (pH 7.8) containing 6 M urea, 2 M thiourea and 2% (w/v) amidosulfobetain-14. Disulphide bridges in proteins were broken by the addition of 45 μg of dithioerythritol prepared fresh in 100 mM Tris buffer, pH 7.8 and left to shake for 60 min. Carbamidomethylation of cysteines was performed by the addition of 108 μg of iodoacetamide prepared fresh in buffer as before, and samples incubated in the dark for 45 min. To reduce urea concentration in the samples to below 1 M 160 μL of Milli-Q water was added to samples. 1 μg of trypsin Gold (Promega, Germany) was added to each sample and incubated at 37 °C for 16 h. Peptides were subject to high pH, low pH fractionation using Isolute 100 mg C18 cartridges (Biotage, Sweden) and eluted into 5 fractions at 6–50% ACN. Peptide eluents were dried by centrifugal evaporation and resuspended at a concentration of 300 ng/μL in 3% ACN, 0.1% FA.

Ultra-definition mass spectrometry (UDMS$^E$) analysis was performed as previously described[44]. Briefly, 0.3 μg of peptides were loaded onto a Waters NanoAquity Liquid Chromatography (LC) system and separated over 60 min on a 75 μm × 150 mm, 1.7 μm Peptide BEH C18 analytical column (Waters, UK), then injected into a Synapt-G2-Si mass spectrometer with Ion Mobility Separation (IMS) (Waters, UK).

Raw data was imported to Progenesis QI version 4.1 (Waters, UK) and each fraction was processed separately before all five fractions were combined into one experiment. MS/MS spectra in the MS raw data files were searched against the UniProt most recent published human reference proteome database with manual addition of porcine trypsin (P00761). Enzyme digestion was set to trypsin, maximum missed cleavage number was set to 3. Cysteine carbamidomethylation was set as a fixed modification, methionine oxidation, protein N-terminal acetylation, glutamine deamidation and protein N-terminal pyrrolidone carboxylation were set as variable modifications. False discovery Rate (FDR) cut off was set at 1% at both peptide and protein level. Protein quantification was performed using unique peptides only. The resulting protein identifications and quantitative data were exported to Excel (Microsoft) for further analysis.

### The immobilisation and the identification of lyso-lipid interacting proteins

Glucosylsphingosine and lyso-Gb3 standards in MeOH were evaporated by centrifugation, then glycosphingolipids were reconstituted in 150 mM phosphate buffered saline (PBS, pH 7.4) to a final concentration of 1 mg/mL. Five milligrams of Dynabeads® M-270 epoxy (Invitrogen, UK) were resuspended in 2 mL of dimethylformamide organic buffer, vortexed and aliquoted into 7 vials (blank, sample and a duplicate) at $3.3 \times 10^8$ beads per vial. Buffer was removed from the beads using a magnet. Beads were washed with 1 mL of 150 mM PBS (pH 7.4), resuspended in 0.1 mL of 100 mM sodium phosphate buffer (pH 7.4) and vortexed. One hundred microlitres of 1 mg/mL lysoGb3, glucosylsphingosine, or PBS (blank) was added to each aliquot of beads. To enhance glycosphingolipid-bead binding 0.1 mL of ammonium sulphate buffer (1 M final concentration) was added. The mixture was incubated at room temperature while mixing on a rotor for 24 h. The beads with bound lyso-glycosphingolipids were placed on a magnet and supernatants removed. The beads were washed with 0.5 mL of 150 mM PBS (pH 7.4) containing 2.5 mg/mL of blocking agent horse myoglobin and further washed in PBS. Lyso-glycosphingolipid bound beads were resuspended in 0.1 mL of 100 mM sodium phosphate buffer (pH 7.4), 0.035 mg/mL protein from SH-SY5Y cell lysate diluted in 150 mM PBS (pH 7.4) added and incubated at room temperature for 1 hour while mixing on the rotor. Supernatants were removed and beads washed with 0.5 mL of 150 mM PBS containing 0.1% Triton-100 (pH 7.4) for 15 min. Triton wash was removed and beads were washed with 150 mM PBS (pH 7.4) twice. Protein-glycosphingolipid bound partners were eluted from the beads by the addition of 40 μL of 100 mM Tris (pH 7.8), containing 6 M urea, 2 M thiourea, 0.2% ABS-14 (w/v), subjected to in-solution digestion without fractionation and LC-MS/MS proteomics analysis[44]. Data were imported to ProteinLynx Global SERVER v3 (Waters, UK), protein identification was performed using UniProt's most recent published human reference

proteome FASTA database (https://www.uniprot.org/proteomes/UP000005640) with manual addition of porcine trypsin (P00761) and horse myoglobin (P68082). Protein quantitative data were exported to Excel (Microsoft) for further analysis. Proteins identified were considered to interact with a glycosphingolipid if they were not detected in the blank – those found to interact with the beads without glycosphingolipid bound to the beads.

### Ubiquitinated protein analysis

Mouse IgG1 antibody that recognises mono and polyubiquitin chains (Clone FK2, Sigma, UK) on proteins was diluted in 200 μL of PBS containing 0.02% of Tween 20 and coupled to protein G magnetic beads (Dynabeads®, Invitrogen, Sigma) at 4 μg antibody/0.5 mg beads overnight while on a rotor at 4 °C. The bead-antibody complex was placed on a magnet, the supernatant removed, the bead-antibody complex washed twice with 200 μL of PBS-Tween 20, then the bead-antibody complex was incubated with 250 μL horse myoglobin (2.5 mg/mL) for 15 min at room temperature while shaking to reduce nonspecific binding. Horse myoglobin was washed off twice with 200 μL of PBS-Tween 20. The bead-antibody complex was incubated for 1 hour at room temperature on the rotor with SH-SY5Y cell lysates prepared as described above. Bead-antibody-antigen complex was washed with 200 μL of PBS-Tween 20 twice, following which antibody-antigen complex was eluted from the beads with 40 μL of digest buffer containing 100 mM Tris (pH 7.8), 6 M urea, 2 M thiourea, 2% (w/v) ASB-14 while rotating for 2 min. Supernatants containing the antibody-antigen complex were subject to proteomic analysis[44]. Identification of proteins present in the FK2 antibody immunocapture was performed as described in the previous section with the only difference that mouse immunoglobulin gamma-1 chain C (P01868) from the commercial antibody was also added to the database. Relative protein quantitation was achieved based on the amount of mouse immunoglobulin gamma-1 chain C peptide added to beads. Data were exported to Microsoft Excel for further analysis.

### Tubulin polymerisation assay

The assay was performed according to the manufacturer's protocol using Cytoskeleton BK011P kit. GlcSph and lyso-Gb3 stocks in DMSO and DMSO buffer control were diluted with Milli-Q water at 10 x desired concentrations. Taxol in DMSO and colchicine were diluted in Milli-Q water at 30 μM. Infinite 200 plate reader (Tecan, Switzerland) was set into fluorescence (excitation 340 nm, emission 435 nm) top reading kinetic mode at 37 °C with a 30 s interval, 30 nm gain and 3 flashes per reading. A black, flat bottom, 96-well pate (Corning, costar, #3686) was placed into the reader for 10 min to warm up. DMSO, taxol, colchicine, and GlcSph and lysoGb3 samples were pipetted in duplicates into the plate and an assay buffer containing final concentration of GTP (1 mM), tubulin (2 mg/mL), glycerol (15%, v/v) was added to each well. The plate was immediately placed into the reader, shaken for 10 s in orbital mode, then fluorescence was recorded for 60 min. The data were exported into Excel (Microsoft Inc) for further processing and visualised using Graghpad, v 10.4 (Prism).

### Analysis of cellular energy charge

Five hundred millilitres of 1 M ice-cold perchloric acid was added to the cells, they were scraped and collected into chilled tubes. Two hundred microliters of 0.5 M Potassium Hydrogen Carbonate in 1 M Potassium Hydroxide were combined with 250 μL of cell extract, vortexed, centrifuged at $14{,}000 \times g$ for 5 min at 4 °C and the supernatant collected into a clean tube. This was stored at −80 °C until derivatisation. One hundred μL of 1 M sodium acetate, pH 4.5 and 10 μL of 4 M chloracetaldehyde were added to 100 μL of cell extract. The mixture was incubated at 60 °C for 40 min and the reaction quenched by putting the sample on ice. Once cool the samples were analysed at 4 °C using High-Performance Liquid Chromatography (HPLC) coupled to fluorescence detector. HPLC equipment consisted of PU-1580 intelligent pumps, AS-950 autosampler and DG-1580-53 in-line mobile phase degasser (Jasco Inc.). Etheno-adenine nucleotides were chromatographically separated over a 150 mm × 4.6 mm, 3 μm ODS C18 analytical

column (Hypersil) using a 100 - 0% 0.2 M potassium phosphate, pH 5 and 0.2 M potassium phosphate with 10% acetonitrile, pH 5 over a 31-min linear gradient at a flow rate of 0.8 mL/min. The FP-920 fluorescence detector (Jasco) was maintained at 290 nm excitation and 415 nm emission wavelength respectively and the data collected with EZChrom Elite v3.17 software (Jasco Inc.). Chromatographic peaks acquired for ATP, ADP and AMP were integrated using elution times of corresponding standards and were used to calculate adenylate energy charge (AEC) according to equation equation[45]:

$$AEC = \frac{[ATP] + \frac{1}{2[ADP]}}{[ATP] + [ADP] + [AMP]}$$

### Analysis of glutathione reductase activity

Cells in PBS were scraped into microcentrifuge tubes, PBS removed by centrifugation for 5 min at 4 °C at $1000 \times g$, 200 μL of ice-cold assay buffer added, the pellets centrifuged for 15 min at $10,000 \times g$ at 4 °C and supernatants collected and stored at $-80$ °C until ready for analysis. The assay was performed using ab83461 GR assay kit (Abcam, UK) according to the manufacturer's instructions. GR reduces GSSG, which in turn reacts with 5,5'-dithiobis-2-nitrobenzoic acid (DTNB) to generate a coloured product (TNB) with strong absorbance at 405 nm. Endogenous GSH was destroyed with 3% hydrogen peroxide (5 μL) added to the sample (100 μL) for 5 min at room temperature, followed by the treatment with catalase (5 μL) for another 5 min. A 0–50 nmol/well TNB standard curve was prepared using a TNB standard and ran alongside the samples, which were added to a 96-well plate (30 μL). A reaction mixture containing GR assay buffer, DTNB, GSSG and NADPH solutions was added to each well (50 μL) and the absorbance read immediately at 405 nm in an Infinite 200 plate reader at room temperature generating an initial reading (A1). The samples were incubated at first for 10 min and the absorbance read again in a kinetic mode with an interval of 1 min for the next hour. TNB standard curve was plotted and a change in absorbance between the initial reading and another time point determined to fall within the linear range of the reaction was applied to the standard curve to interpolate the amount of TNB in each sample. The activity of GR was calculated as follows: GR activity = (ΔB*Sample dilution factor)/((T2−T1)*0.9*V) (nmol/min/mL), where ΔB (nmol) is the amount of TNB interpolated from the standard curve, T1 (min) is the time of the initial reading, T2 (min) is the time of the second reading, V (mL) is the sample volume and 0.9 is the sample volume change factor. The resulting value was normalised to total protein determined by the Bradford method to yield GR activity (nmol/min/mg).

### Analysis of malate dehydrogenase activity

Cells were collected by scaping. PBS was removed by centrifugation for 5 min at 4 °C at $1000 \times g$ and pellets homogenised in 100 μL of ice-cold assay extraction buffer for 10 min on ice. The pellets were centrifuged at $10,000 \times g$ for 5 min at 4 °C, the supernatants collected and stored at $-80$ °C until ready for analysis. The assay was performed using an MDH assay kit (Abcam, ab183305, UK) according to the manufacturer's instructions. One hundred microlitres of ice-cold assay buffer was added to each pellet, placed on ice for 10 min and spun at $10,000 \times g$ for 5 min at 4 °C, the supernatants collected and diluted 1 in 10 in the assay buffer. Ten microlitres of each sample dilution was aliquoted out for total protein measurement by the Bradford method. Samples were analysed in duplicates. A 6-point calibration curve containing 0–12.5 nmol of NADH standard was prepared in duplicates and sample blanks for each sample dilution was run alongside. The reaction mixture for the samples and standards contained samples/standards, assay buffer, enzyme mix, developer and substrate. Sample blanks had all the above except for the substrate. After an initial shake the plate was incubated at 37 °C in Infinite 200 plate reader, the initial absorbance measured at 450 nm, then the absorbance was measured in a kinetic cycle for 30 min every 5 min until the value of the most active sample was greater than the value of the highest standard. The activity of MDH was calculated

as follows: MDH activity $= \frac{Sa}{(\text{Reaction Time})x \, Sv}$ where Sa is the amount of NADH (nmol) generated in the sample between the final and initial time of reading (interpolated from the standard curve) Reaction Time is the difference between the final and initial time of reading (min) Sv is the sample volume used in the rection (mL). The resulting value was normalised to total protein to yield MDH activity (nmol/min/mg).

### Analysis of mitochondrial malate dehydrogenase activity

Samples were prepared in the same manner as for MDH assay with the exception that the pellets were incubated in 200 μL of assay extraction buffer for 20 min on ice. The sample protein was determined by the Bradford assay and each sample was diluted to contain 150 μg of protein. The assay was performed using MDH2 activity kit (ab119693, Abcam, UK) according to the manufacturer's protocol. Samples were diluted with incubation buffer and together with a dilution series of a control sample (100 μL) were added to the wells to capture mitochondrial MDH2 and incubated at room temperature while shaking at 300 rpm for 3 h. The buffer was removed, samples washed with blocking buffer and the wash repeated. An activity solution (100 μL) containing sodium malate, $NAD^+$, coupler, reagent dye and assay buffer giving the final concentration of malate and $NAD^+$ 5 mM and 4 mM respectively was added to the samples, absorbance read at 450 nm in a kinetic cycle for 25 min with a 20 s interval with shaking before and between readings. MDH2 activity was calculated using the Beer-Lambert law and the dye extinction coefficient was 37 $mM^{-1} \, cm^{-1}$.

### Analysis of citrate synthase activity

CS activity was determined by a modified method based on the premise that upon the action of CS Coenzyme A and citrate are liberated and Coenzyme A reacts with DTNB, which causes a colorimetric change and can be monitored at 412 nm[38]. Cells were collected on ice and centrifuged at 4 °C, the PBS removed and the pellet re-suspended in 500 μL of 10 mM Tris buffer, pH 7.4 containing 320 mM sucrose and 1 mM EDTA. Each pellet was split into two 250 μL aliquots, one for the CS the other for complex I activity assays, and quickly snap frozen on dry ice before storage at $-80$ °C prior to analyses. Cell lysates were thawed and snap frozen twice using ethanol and dry ice and each sample cuvette was prepared to contain 950 μL of 100 mM Tris/ 0.1% Triton, (v/v), pH 8.0, 20 μL of cell homogenate, 10 μL of 10 mM acetyl-CoA and 10 μL of 20 mM DTNB. The corresponding reference cuvettes were prepared in the same way but had 960 μL of 100 mM Tris/0.1% Triton, (v/v). The cuvettes were covered with parafilm, gently inverted twice before they were put into a Kontron Uvikon 941 spectrophotometer (Nothstar Scientific) maintained at 30 °C. Ten microlitres of 20 mM oxaloacetate was added to the sample cuvettes only, gently mixed, and the reaction monitored at 412 nm for 30 min and measured every 30 s. CS activity was calculated using the Beer-Lambert Law, where path length was 1 cm, total volume was 1 mL and the extinction coefficient of DTNB was 13.6 $mM^{-1} \, cm^{-1}$. CS activity was expressed in nmol/min/mg.

### Analysis of mitochondrial complex I activity

A modified method of Ragan et al. was used to determine complex I activity[46]. Each sample was prepared to contain 800 μL of 25 mM phosphate buffer containing 10 mM magnesium chloride, pH 7.2, 10 μL of 100 mM potassium cyanide, 50 μL of 50 mM BSA, 30 μL of 5 mM NADH, 20 μL of cell homogenate and 80 μL of water. The corresponding reference cuvettes were prepared in the same way but contained 90 μL of water. The cuvettes covered with parafilm were gently inverted twice before being put into a Kontron Uvikon 941 spectrophotometer maintained at 30 °C. Ten microlitres of Coenzyme $Q_1$ was added to the sample cuvettes only, gently mixed and left to reach 30 °C, and the reaction measured at 340 nm for 5 min at 30 s intervals. After 5 min elapsed 20 μL of 1 mM rotenone was added to the sample cuvettes only and after a 2–3 min wait the measurement continued for a further 5 min. Using the Beer-Lambert Law the concentration of oxidised NADH was calculated where the change in absorbance at 340 nm after rotenone addition was subtracted from the change in absorbance before rotenone addition, the extinction coefficient of NADH was

$6.22\ \mathrm{mM^{-1}\ cm^{-1}}$, path length was 1 cm and volume was 1 mL. The resulting value expressed as nmol/min/mg was divided by the value of CS activity to obtain a ratio.

## Analysis of mitochondrial complex II/III activity

The activity of complex II/III was assayed based on the King method[47]. The oxidation of succinate to fumarate by complex II shuttling electrons from Coenzyme $QH_2$ to reduce cytochrome *c* by complex III and is measured spectrophotometrically at 550 nm as a succinate dependent antimycin A (AA) sensitive reduction of cytochrome *c*. Each sample cuvette was prepared to contain a final concentration of 1 mM potassium cyanide, 300 µM EDTA, 100 µM cytochrome *c*, in a 166 mM potassium phosphate buffer pH 7.4 with addition of 185 µL of water. The corresponding reference cuvette was prepared in the same manner but 225 µL of water was added. 20 µL of sample was then added to each sample cuvette only, gently mixed and left to warm to 30 °C in the spectrophotometer. The reaction was initiated by the addition of 40 µL of 0.5 mM succinate into the sample cuvettes and the absorbance followed for 5 min. The absorbance for each sample was determined by subtracting the difference in absorbance prior to and after the addition of AA and Beer-Lambert law was applied using extinction coefficient of cytochrome *c* as $19.2 \times 10^3\ \mathrm{M^{-1}\ cm^{-1}}$. The resulting values were normalised to the activity of CS and expressed as ratios.

## Analysis of alpha enolase activity

The activity was examined using an Abcam ab117994 kit according to manufacturer's protocol. Cells were washed with ice-cold PBS and collected on ice by scraping. PBS was removed by centrifugation at $500 \times g$ for 5 min at 4 °C and aspiration, pellets washed with 100 µL ice-cold PBS, centrifugation repeated and PBS removed. Pellets were dissolved in 200 µL of the kit extraction buffer while incubated on ice for 20 min, spun at $16,000 \times g$ at 4 °C for 20 min, supernatants collected, total protein determined by the Bradford method and the supernatants stored at −80 °C till ready for analysis. Samples were diluted with incubation buffer to contain 150 µg of protein and a control sample dilution series from 0 to 250 µg of total protein was prepared. The native ENO1 was immunocaptured from the samples during a 2-h incubation at room temperature while shaking at 300 rpm. The buffer was removed, samples washed with blocking buffer and the wash repeated. An activity solution (100 µL), containing 0.5 mM NADH, 0.25 mM 2DG, PK, LDH and assay buffer was added to the samples and absorbance was immediately recorded at 340 nm over a period of 40 min with an interval of 20 s at 37 °C in an Infinite f200 plate reader. ENO1 activity was determined as the change in absorbance per min per µg of protein being proportionate to the consumption of NADH in the reaction mixture by the Beer-Lambert Law using the extinction coefficient of NADH as $6.22\ \mathrm{mM^{-1}\ cm^{-1}}$.

## Analysis of ATP production and glycolytic rates using Agilent XFp Seahorse analyser

**Sample preparation.** XFp cartridges were left hydrating overnight prior to the day of the assay in MilliQ water. Alamar blue cell viability reagent was added to each well (1/10th volume) and left to permeate the cells overnight. On the day of the analysis cell viability was assessed by measuring colour change and fluorescence of the dye (excitation 530 nm, emission 590 nm) using an Infinite f200 plate reader (Tecan, Switzerland). Fresh XF DMEM assay medium pH 7.4 was prepared with 1 mM sodium pyruvate, 2 mM glutamine and 10 mM glucose on the day of the assay. Following the 72-h GlcSph treatment cell culture medium was carefully removed and cells checked under the microscope for any perturbations. They were then washed with XF medium, 180 µL of assay medium was added to each well, the plate checked under the microscope again and incubated for 1 h at 37 °C with no $CO_2$ for 45–60 min to remove $CO_2$. After degassing cells were washed with the assay medium, checked under the microscope and the plate placed in the Seahorse analyser for the analysis.

**ATP production rate.** Port A of the XFp cartridge was designated for acute treatment of both control and lyso-GSL treated cells with 3 µM UK5099 (mitochondrial pyruvate shuttle inhibitor). Manufacturer's protocol was followed using ATP rate assay kit where port B contained 1.5 µM oligomycin, port C contained 0.5 µM rotenone/antimycin A. Data were exported into Seahorse Wave software (v 2.6.1, Agilent), normalised to cell viability and an ATP rate report was generated.

**Glycolytic rate.** Port A contained 0.5 µM rotenone/antimycin A and port B contained 500 µM 2-deoxy-D-glucose – an allosteric hexokinase inhibitor for glycolysis inhibition. The data were exported into Seahorse Wave software (v 2.6.1, Agilent), normalised to cell viability and a glycolytic rate report was generated.

**Statistics and reproducibility.** Data from the ATP and glycolytic rate reports and functional assays were exported into Graphpad Prism software (V9, USA) and analysed by a 2-way ANOVA with a Sidak's post-hoc multiple comparison test for the ATP production and glycolytic rates, and a Kruskal–Wallis test with a Dunn's post-hoc test for the activity of mitochondrial complexes I-III, citrate synthase, malate dehydrogenases, glutathione reductase, alpha enolase and cellular energy charge assays. Results were considered to be statistically significant with $p$ value < 0.05.

Proteomics data were analysed for pathway enrichment using IPA (QIAGEN Inc, https://digitalinsights.qiagen.com/products-overview/discovery-insights-portfolio/analysis-and-visualization/qiagen-ipa/). Input variables were set to proteins that demonstrated a significantly altered expression between the control and lyso-lipid treated cells, with fold-change as expression observation. The output pathways were set to those with $p < 0.05$. Gene ontology annotations were extracted using PANTHER bioinformatics tool (https://www.pantherdb.org/).

## Reporting summary

Further information on research design is available in the Nature Portfolio Reporting Summary linked to this article.

## Data availability

The mass spectrometry proteomics data have been deposited to the ProteomeXchange Consortium via the PRIDE partner repository with the dataset identifier PXD061666 and 10.6019/PXD061666. Source data for all data represented in graphs within the figures are provided in a Supplementary Data file. Supplementary data are provided with this paper.

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

## Acknowledgements

This study has been part funded by the UCL Biological Mass Spectrometry Centre. All research at Great Ormond Street Hospital NHS Foundation Trust and UCL Great Ormond Street Institute of Child Health is made possible by the NIHR Great Ormond Street Hospital Biomedical Research Centre. The

views expressed are those of the author(s) and not necessarily those of the NHS, the NIHR or the Department of Health. We would also like to thank the Peto Foundation for their generous donations.

## Author contributions

V.N. designed and performed experiments, analysed and interpreted data, wrote the original draft and revised the manuscript. R.V., S.E., J.H., and T.B. provided support for data collection, analysis and revised the manuscript. W.H. conceptualised and funded the study, W.H. and K.M. designed the experiments, supervised research and revised the manuscript. All authors read and approved the final version of the manuscript.

## Competing interests

The authors declare no competing interests.
