## [Transparent Peer Review file · Communications Biology]

Glucosylsphingosine affects mitochondrial function in a neuronal cell model

Corresponding Author: Dr Wendy Heywood

Version 0:

Reviewer comments:

Reviewer #1

(Remarks to the Author)

The manuscript COMMSBIO-25-2882-T, 'Glucosylsphingosine affects mitochondrial function in a neuronal cell model', reports an investigation of the potential pathogenic role of glucosylsphingosine (GlcSph), a secondary metabolite of glucosylceramide known to accumulate in Gaucher disease (GD) patients deficient in GCase degrading GlcCer to Cer in lysosomes.

It was observed by proteomics analysis that incubation of SH-Sy5y cells with GlcSph at plasma concentrations occurring in GD led to effects on the TCA cycle, mitochondria, glycolysis and protein ubiquitination. Of interest, GlcSph was also found to reduce ATP production, cause oxidative stress and increase glycolysis. Disturbed cellular energy metabolism during GCase deficiency (Gaucher disease) is already well documented; the present manuscript firstly suggests a direct role for GlcSph in this, although (surprisingly) no inhibitory effect of GlcSph on complex I was noted.

Analysis of proteins binding to (immobilised modified) GlcSph revealed binding affinity of GlcSph for tubulin alpha. Its binding induced a specific increase of ubiquitination of α and β tubulins, suggesting that GlcSph potentially affects cellular transport.

The manuscript is of high quality and the authors are applauded with their original approaches and related findings. Specific comments.

1. In the manuscript use is made of immobilized GlcSph to Dynobeads. It is not specifically mentioned, but presumably the immobilisation requires the reaction of the free amine in GlcSph to a reactive group the beads. How does such immobilization impact on the features of the lyso-lipid? It no longer contains its characteristic (charged) amine, how is the sphingosine moiety sterically affected?

Would a competition of binding experiment using an excess of free GlcSph (monitoring which proteins are no longer detected with proteomics) be useful to validate binding of natural (free) GlcSph?

The authors are asked for a comment.

2. At lines 302-306, the authors report a noted shift toward glycolysis and reduction in the ATP reduction in dopaminergic neurons in mice with PD. Earlier impaired complex I activity has been reported in GD-derived cells. See: Osellame LD, Rahim AA, Hargreaves IP, Gegg ME, Richard-Londt A, Brandner S, Waddington SN, Schapira AHV, Duchon MR. Mitochondria and quality control defects in a mouse model of Gaucher disease--links to Parkinson's disease. *Cell Metab.* 2013 Jun 4;17(6):941-953. The manuscript reports no evidence for inhibition by GlcSph of complex I activity as measured in homogenates of SH-Sy5y cells. The authors mention (line 304) that further investigations are necessary to determine whether GlcSph truly does not affect the activity of complexes I-III in other more vulnerable cell types such as dopaminergic neurons that are more reliant on OXPHOS.

One wonders what the outcome would be of exposing, for different time periods, intact mitochondria to GlcSph: this pre-incubation would allow conversion of GlcSph to an complex I inhibitory glucolipid (e.g. GlcCer by re-acylation?). The authors are asked for a comment on this thought.

Reviewer #2

(Remarks to the Author)

The authors conducted a proteomics analysis to investigate the molecular mechanisms of glucosylsphingosine, a substrate that accumulates in Gaucher disease, affects cultured neuronal cells. Following functional analyses confirmed connections

with mitochondrial dysfunction and altered ubiquitination of tubulin. These findings, along with their potential relevance to Gaucher disease and Parkinson's disease, are highly interesting. However, the study does not present direct evidence linking the results specifically to Gaucher disease. Including such evidence could further strength this manuscript. Additionally, as the authors mentioned that the endogenous accumulation of glucosylsphingosine observed in Gaucher disease cells occurs within lysosomes due to a deficiency in degradative enzyme. Therefore, it remains unclear to what extent the effects of exogenously applied glucosylsphingosine, used in this study, are related to the pathophysiology of Gaucher disease. For instance, while the authors showed that exogenous glucosylsphingosine was taken up by SH-SY5Y cells, it is unclear where it localizes within the cells after uptake and to what extent it accumulated within lysosomes. Consequently, questions remain regarding how closely the findings from this model correlate with actual Gaucher disease cells.

Therefore, it would be necessary to conduct verification experiments as additional data on these molecular pathologies using GBA1-deficient neuronal cell models, or brain tissues from model animals. Furthermore, by adding complementary experiments with GBA1 gene expression, it would possibly insist on the direct relevance of this study to Gaucher disease and Parkinson disease as well.

Reviewer #3

(Remarks to the Author)

In the current submission, Dr. Heywood et al investigated the effects of glucosylsphingosine (GlcSph) on cellular proteomics in SH-Sy5y cells and identified broad effects on the TCA cycle, mitochondrial function, glycolysis, and protein ubiquitination. Evaluating mitochondrial functions using Seahorse assays further confirmed reduced ATP production, excessive oxidative stress, and upregulated glycolysis as a compensation. In an in vitro assay where cell lysates were incubated with immobilized GlcSph, GlcSph was demonstrated to bind to α tubulin and to increase α and β tubulin ubiquitination. Overall, the experiments were carried out nicely and the findings are interesting, suggesting that elevated GlcSph could impact various aspects of mitochondrial functions.

Specific comments:

It is understandable that GlcSph was immobilized to pull down and identify interacting proteins in cell lysates. However, protein ubiquitination elicited by GlcSph treatments could have been carried out after treating live cells with GlcSph, followed by lysis and ubiquitination antibody pull down. This way protein ubiquitination might occur more physiologically.

Version 1:

Reviewer comments:

Reviewer #1

(Remarks to the Author)

The questions raised are appropriately addressed and satisfactorily answered.

Reviewer #2

(Remarks to the Author)

The supplementation of exogenous GlcSph is still too artificial to discuss the relevance of findings obtained from such experiments to the molecular pathogenesis of Gaucher disease.

Therefore, in addition to using normal SH-SY-Y cells, it would be feasible to utilize cells in which GBA1 has been knocked down either by siRNA or disrupted by genome editing techniques. By confirming phenomena such as mitochondrial dysfunction and enhanced ubiquitination of tubulin in GBA1-deficient neuronal cells, the authors' findings could be more strongly substantiated.

Reviewer #3

(Remarks to the Author)

Thank you for the clarification on the experiment workflow!

Reviewers' comments:

Reviewer #1 (Remarks to the Author):

The manuscript COMMSBIO-25-2882-T, 'Glucosylsphingosine affects mitochondrial function in a neuronal cell model', reports an investigation of the potential pathogenic role of glucosylsphingosine (GlcSph), a secondary metabolite of glucosylceramide known to accumulate in Gaucher disease (GD) patients deficient in GCase degrading GlcCer to Cer in lysosomes.

It was observed by proteomics analysis that incubation of SH-Sy5y cells with GlcSph at plasma concentrations occurring in GD led to effects on the TCA cycle, mitochondria, glycolysis and protein ubiquitination. Of interest, GlcSph was also found to reduce ATP production, cause oxidative stress and increase glycolysis. Disturbed cellular energy metabolism during GCase deficiency (Gaucher disease) is already well documented; the present manuscript firstly suggests a direct role for GlcSph in this, although (surprisingly) no inhibitory effect of GlcSph on complex I was noted. Analysis of proteins binding to (immobilised modified) GlcSph revealed binding affinity of GlcSph for tubulin alpha. Its binding induced a specific increase of ubiquitination of α and β tubulins, suggesting that GlcSph potentially affects cellular transport.

The manuscript is of high quality and the authors are applauded with their original approaches and related findings.

Specific comments.

1. In the manuscript use is made of immobilized GlcSph to Dynobeads. It is not specifically mentioned, but presumably the immobilisation requires the reaction of the free amine in GlcSph to a reactive group the beads. How does such immobilization impact on the features of the lyso-lipid? It no longer contains its characteristic (charged) amine, how is the sphingosine moiety sterically affected?

Yes, we expect it is the amine group that links to the epoxy group on the bead, which may limit the binding of proteins to the lyso-lipids without the free amine group available. We did investigate the use of an antibody to GlcSph to immobilise the lipid although it is not known which functional group of the lipid the antibody binds to, it was not very good and introduced the need to control for proteins that bound to the antibody.

However, the effects and associated disease phenotypes of the two lipids are so different we suspect that the differences are largely caused by the different sugar groups and not just the common amine group that they both have. Hence why a comparison between these two lyso-lipids was considered the best way to look for specific effects of GlcSph. We have added a sentence (page 10) in the results addressing the limitation of the immobilised lipid experiment. "Of note, immobilisation of lyso-lipids is expected to obscure their free amine group as it reacts with the epoxy group on magnetic beads. This may have affected the proteins immobilised GlcSph bound to, and is a limitation of this study. However, the effects on the cell proteome and associated disease phenotypes of the two lyso-lipids are so different we suspect that the differences are largely caused by the different sugar moieties and not just the common amine group that both lyso-lipids have. Hence why a comparison with blank beads and lyso-Gb3 was considered the best way to look for specific effects of GlcSph."

Would a competition of binding experiment using an excess of free GlcSph (monitoring which proteins are no longer detected with proteomics) be useful to validate binding of natural (free) GlcSph?

We thank the reviewer for this clever suggestion, however this would only help confirm those proteins binding to free GlcSph but would still miss any that are specific only to free GlcSph or specifically binding to the amine group.

The authors are asked for a comment.

2. At lines 302-306, the authors report a noted shift toward glycolysis and reduction in the ATP reduction in dopaminergic neurons in mice with PD. Earlier impaired complex I activity has been reported in GD-derived cells. See: Osellame LD, Rahim AA, Hargreaves IP, Gegg ME, Richard-Londt A, Brandner S, Waddington SN, Schapira AHV, Duchen MR. Mitochondria and quality control defects in a mouse model of Gaucher disease--links to Parkinson's disease. *Cell Metab.* 2013 Jun 4;17(6):941-953. The manuscript reports no evidence for inhibition by GlcSph of complex I activity as measured in homogenates of SH-Sy5y cells. The authors mention (line 304) that further investigations are necessary to determine whether GlcSph truly does not affect the activity of complexes I-III in other more vulnerable cell types such as dopaminergic neurons that are more reliant on OXPHOS. One wonders what the outcome would be of exposing, for different time periods, intact mitochondria to GlcSph: this pre-incubation would allow conversion of GlcSph to an complex I inhibitory glucolipid (e.g. GlcCer by re-acylation?). The authors are asked for a comment on this thought.

Mitochondria will not operate efficiently in isolation for long enough to delineate any effect from incubation with GlcSph. Understanding the effect GlcSph has on mitochondria whilst in the cell may have been inconclusive in our study but we believe this was the appropriate way as it may not be a direct consequence of the GlcSph, as GlcSph may have affected other cellular pathways that in turn affect mitochondria. Also, we have no way of confirming that GlcSph directly encounters mitochondria in the cell (see response to reviewer 2 comments) to confirm that it's a direct mechanism.

Regarding reacylation – this is unlikely. In the glycosphingolipid (GSL) degradation pathway GSLs are sequentially degraded by the removal of terminal sugars one at a time to sphingosine and a free fatty acid. Although acylation of sphingosine is a naturally occurring process in the GSL anabolism, it is not known whether acylation of a sugar moiety-bound sphingosine (lysolipid) is. We suspect if that was the case the lysolipid biomarkers of the glycosphingolipidoses would not be accumulating in the first place. Some modifications of the lysolipids do take place as is well documented by Prof Christiane Aurey Blais's work on lyso-Gb3 analogues. She describes modifications of the sphingosine but these are not acyl chains. See "Globotriaosylsphingosine (lyso-Gb3) and analogues in plasma and urine of patients with Fabry disease and correlations with long-term treatment and genotypes in a nationwide female Danish cohort". Grigoris Effraimidis et al., *J Med Genet.* 2021 Oct;58(10):692-700. These modifications, which some are suspected to be methylation, make the lyso-lipid more hydrophilic so it is easier to excrete hence their higher abundance in urine. They have not been as well described for GlcSph but we have observed them in our analyses in GD patients (unpublished) and are confident a similar mechanism exists.

Reviewer #2 (Remarks to the Author):

The authors conducted a proteomics analysis to investigate the molecular mechanisms of glucosylsphingosine, a substrate that accumulates in Gaucher disease, affects cultured neuronal cells. Following functional analyses confirmed connections with mitochondrial dysfunction and altered

ubiquitination of tubulin. These findings, along with their potential relevance to Gaucher disease and Parkinson's disease, are highly interesting. However, the study does not present direct evidence linking the results specifically to Gaucher disease. Including such evidence could further strength this manuscript.

Additionally, as the authors mentioned that the endogenous accumulation of glucosylsphingosine observed in Gaucher disease cells occurs within lysosomes due to a deficiency in degradative enzyme. Therefore, it remains unclear to what extent the effects of exogenously applied glucosylsphingosine, used in this study, are related to the pathophysiology of Gaucher disease. For instance, while the authors showed that exogenous glucosylsphingosine was taken up by SH-SY5Y cells, it is unclear where it localizes within the cells after uptake and to what extent it accumulated within lysosomes. Consequently, questions remain regarding how closely the findings from this model correlate with actual Gaucher disease cells.

We acknowledge a limitation of this study is not assessing the toxicity of only lysosomal derived GlcSph. To do so a way to block exocytosis of GlcSph would be required, which would profoundly affect the cell in other ways therefore this is not a feasible experiment. We have only been able to assess the effect of exogenous GlcSph, however, this is still highly relevant to GD. GlcSph is taken up by cells from circulation and as it cannot be degraded it accumulates. We did demonstrate in supplementary information a dose dependent uptake and accumulation of GlcSph thereby confirming it does accumulate, we just can't confirm if it accumulates in the lysosome. Furthermore, is not actually possible to achieve tracking of GlcSph within the cell. There are no good antibodies for GlcSph to conduct imaging microscopy studies and the BODIPY tags available are for GlcCer not GlcSph. Subcellular fractionation analysis as an alternative can only achieve subcellular enrichment and is therefore limited, and would still not entirely answer the question. Whilst it would be interesting to see where GlcSph goes within the cell this is not essential to our study as we clearly demonstrate there is a toxic effect on mitochondria.

Therefore, it would be necessary to conduct verification experiments as additional data on these molecular pathologies using GBA1-deficient neuronal cell models, or brain tissues from model animals. Furthermore, by adding complementary experiments with GBA1 gene expression, it would possibly insist on the direct relevance of this study to Gaucher disease and Parkinson disease as well.

The aim of this work was to determine the specific effects of GlcSph, which is highly relevant to Gaucher disease. Many other studies have demonstrated GlcSph toxicity, for example, we cited Lukas, J. *et al.* "Glucosylsphingosine causes hematological and visceral changes in mice—evidence for a pathophysiological role in gaucher disease". *Int J Mol Sci* **18**, (2017), which nicely demonstrated the relevance of exogenous GlcSph to pathophysiology of Gaucher disease – it causes the same haematological and visceral changes in mice as those observed in GD patients. Therefore, it is important to understand GlcSph specific effects and homeostasis, which could identify therapeutic targets.

It is not possible to delineate these specific effects in GD models where other GD disease processes would confound observations, i.e. lysosomal hypertrophy from GlcCer accumulation. The work of Prof Johannes Aerts in zebrafish, which we cited, has nicely addressed specific effects of GlcSph and

GlcCer accumulation by using acid ceramidase silencing (see Lelieveld, L. *et al.* "Consequences of excessive glucosylsphingosine in glucocerebrosidase-deficient zebrafish", *J of Lipid Research* **5**, (2022)). By using a GD model we would only be repeating this work. We believe our work advances on the work of Prof Aerts by identifying the specific molecular mechanisms of GlcSph toxicity and explains the cellular vulnerability observed in his work.

Reviewer #3 (Remarks to the Author):

In the current submission, Dr. Heywood et al investigated the effects of glucosylsphingosine (GlcSph) on cellular proteomics in SH-Sy5y cells and identified broad effects on the TCA cycle, mitochondrial function, glycolysis, and protein ubiquitination. Evaluating mitochondrial functions using Seahorse assays further confirmed reduced ATP production, excessive oxidative stress, and upregulated glycolysis as a compensation. In an in vitro assay where cell lysates were incubated with immobilized GlcSph, GlcSph was demonstrated to bind to α tubulin and to increase α and β tubulin ubiquitination. Overall, the experiments were carried out nicely and the findings are interesting, suggesting that elevated GlcSph could impact various aspects of mitochondrial functions.

Specific comments:

It is understandable that GlcSph was immobilized to pull down and identify interacting proteins in cell lysates. However, protein ubiquitination elicited by GlcSph treatments could have been carried out after treating live cells with GlcSph, followed by lysis and ubiquitination antibody pull down. This way protein ubiquitination might occur more physiologically.

We would like to reassure the reviewer that this was what we did do. To clarify there were two bead-based experiments. One where GlcSph was immobilised using magnetic beads and incubated with untreated cells to identify what proteins it interacts with, and the other, where ubiquitinated proteins were identified using lysates of live cells **pre-treated** with lysolipids, and an antibody against mono- and poly-ubiquitin chains on proteins (see manuscript methods).